# Neurexins regulate presynaptic GABA$_B$-receptors at central synapses

Fujun Luo [1,2,3 ✉], Alessandra Sclip [1,3], Sean Merrill[1] & Thomas C. Südhof [1]

Diverse signaling complexes are precisely assembled at the presynaptic active zone for dynamic modulation of synaptic transmission and synaptic plasticity. Presynaptic GABA$_B$-receptors nucleate critical signaling complexes regulating neurotransmitter release at most synapses. However, the molecular mechanisms underlying assembly of GABA$_B$-receptor signaling complexes remain unclear. Here we show that neurexins are required for the localization and function of presynaptic GABA$_B$-receptor signaling complexes. At four model synapses, excitatory calyx of Held synapses in the brainstem, excitatory and inhibitory synapses on hippocampal CA1-region pyramidal neurons, and inhibitory basket cell synapses in the cerebellum, deletion of neurexins rendered neurotransmitter release significantly less sensitive to GABA$_B$-receptor activation. Moreover, deletion of neurexins caused a loss of GABA$_B$-receptors from the presynaptic active zone of the calyx synapse. These findings extend the role of neurexins at the presynaptic active zone to enabling GABA$_B$-receptor signaling, supporting the notion that neurexins function as central organizers of active zone signaling complexes.

[1] Department of Molecular and Cellular Physiology and Howard Hughes Medical Institute, Stanford University Medical School, Stanford, CA, USA. [2] Bioland Laboratory (Guangzhou Regenerative Medicine and Health-Guangdong Laboratory), Guangzhou, Guangdong, China. [3]These authors contributed equally: Fujun Luo, Alessandra Sclip. ✉email: luo_fujun@grmh-gdl.cn

Action potential-evoked neurotransmitter release occurs with high speed and precision at the presynaptic active zone, which is tightly organized at the nanometer level in the nerve terminal. Four key functions of the presynaptic active zone have been proposed: tethering synaptic vesicles to release sites, priming synaptic vesicles for rapid $Ca^{2+}$-triggered fusion, clustering voltage-gated $Ca^{2+}$-channels to release sites, and coordinating the trans-synaptic alignment of presynaptic and postsynaptic signaling complexes[1]. Studies have revealed that every aspect of active zone functions is mediated by evolutionarily conserved scaffolding molecules, including RIMs, RIM-binding proteins, ELKS/Bruchpilot, and Munc13[2–8]. These molecules interact with each other and other signaling proteins to determine the number of $Ca^{2+}$-channels and their spatial coupling with primed synaptic vesicles, both of which are critical for determining the release probability of a synapse[2,3,6,8]. Moreover, the expression, distribution, and functional properties of $Ca^{2+}$-channels at the presynaptic active zone are extensively modulated by diverse G-protein coupled receptors (GPCRs)[9–11]. This modulation greatly enhances the power of synaptic computations in various forms of short-term and long-term plasticity[12].

$GABA_B$-receptors are GPCRs activated by GABA, the main inhibitory neurotransmitter in the mammalian brain. $GABA_B$-receptors are widely expressed in both presynaptic and postsynaptic compartments of almost all neurons as well as in astrocytes[10,13], and play important roles in regulating synaptic function and short-term plasticity[14]. In particular, activation of presynaptic $GABA_B$-receptors prominently inhibits neurotransmitter release at excitatory and inhibitory synapses by suppressing the activity of $Ca^{2+}$-channels[15–19], and by additional mechanisms independent of $Ca^{2+}$-channels[20–23].

$GABA_B$-receptors are assembled as heterodimers of two principal subunits, $GABA_{B1}$ and $GABA_{B2}$. Both subunits are essential for the formation of functional receptors[14]. Two $GABA_{B1}$ isoforms are generated from distinct promoters, $GABA_{B1a}$ and $GABA_{B1b}$, that contain or lack, respectively, N-terminal "sushi" domains[24]. $GABA_{B1a}$-receptors are preferentially targeted to presynaptic terminals via interactions mediated by these sushi domains[24,25]. $GABA_B$-receptor complexes contain additional proteins that may regulate their pharmacological and kinetic properties[26], and control their surface expression and stability[27,28]. Quantitative proteomics of the macromolecular composition of native $GABA_B$-receptor complexes identified multiple interacting proteins, including $Ca^{2+}$-channels, APP, and calsyntenin, suggesting that these proteins' association with $GABA_B$-receptors may increase the diversity of $GABA_B$-receptor signaling[26]. For example, it has been shown that the secreted cleaved APPs fragment of APP binds directly to the $GABA_{B1a}$ sushi domain to regulate synaptic transmission, and short-term facilitation in mouse hippocampal synapses[29]. Additionally, APP may associate with JIP and calsyntenin to promote axonal trafficking of $GABA_B$-receptors[30]. Interestingly, quantitative proteomic analysis of the molecular composition of $Ca^{2+}$-channel complexes also suggested that $GABA_B$-receptors interact with $Ca^{2+}$-channels[31]. Thus, $GABA_B$-receptors, G-proteins, and $Ca^{2+}$-channels may act as a single signaling complex in presynaptic terminals[16]. However, it remains unclear how the complex is formed and regulated.

Neurexins are evolutionarily conserved cell adhesion molecules that play key roles in shaping the properties of synapses[32]. A recent study has uncovered that conditional deletions of all neurexins in different synapses produce severe but dramatically different phenotypes, suggesting that neurexins may act as synapse-specific functional organizers instead of playing a canonical role in all synapses[33]. Further, pan-neurexin ablation at the calyx of Held, a model synapse allowing precise characterization of synaptic properties, showed that neurexins are crucial

organizers for the clustering of $Ca^{2+}$-channels at the active zone and their tight coupling to the release machinery and to BK-channels that modulate presynaptic action potentials[34]. However, it remains unclear whether neurexins organize the general assembly of active zone signaling complexes. Here, we addressed this question by testing whether $GABA_B$-receptor-mediated modulation of presynaptic $Ca^{2+}$-channels, and neurotransmitter release is impacted after deletion of all neurexins. We show that at the calyx of Held synapse, deletion of all neurexins strongly impaired $GABA_B$-receptor mediated modulation of neurotransmitter release. Interestingly, we replicated these effects in three other central synapses, including excitatory and inhibitory synapses formed on CA1 pyramidal neurons in the hippocampus, and inhibitory synapses formed on Purkinje cells of the cerebellum. These data suggest that neurexins universally regulate presynaptic $GABA_B$-receptor signaling at various central synapses.

## Results

**Neurexins are required for $GABA_B$-receptor function at the calyx of Held.** To analyze the potential role of neurexins in organizing presynaptic $GABA_B$-receptors, we first studied the calyx of Held, a giant glutamatergic synapse in the medial nucleus of the trapezoid body (MNTB)[35–37]. At calyx synapses, presynaptic terminals wrap around the soma of principal MNTB neurons and form 500–600 synaptic contacts. Their large size makes it possible to patch calyx terminals, enabling direct access to their presynaptic cytosol (Fig. 1a). We crossed triple Nrxn123 conditional KO mice with Pv-Cre mice to delete all neurexins from parvalbumin-positive ($Pv^+$) neurons, including the presynaptic neurons of the calyx of Held synapse[33] (Fig. 1b). We then analyzed littermate triple Nrxn123 conditional KO mice lacking (referred to as "control") or containing the Pv-Cre allele (referred to as Nrxn123 TKO mice).

Previous studies revealed that $GABA_B$-receptors play an important role in controlling the release probability at the calyx of Held synapse[18,38]. Consistent with this observation, activating $GABA_B$-receptors with the specific and potent agonist SKF 97541 (SKF) strongly decreased evoked EPSCs by 80% in calyx synapses of control mice (Fig. 1c, d). Moreover, SKF increased the paired-pulse ratio (PPR), consistent with a reduced release probability (Fig. 1f). The effect of SKF was reversible and could be completely prevented by pre-incubation of calyx synapses with the $GABA_B$-receptor antagonist CGP 55845 (Supplementary Fig. 1a, b). SKF had no effect on the amplitude, frequency, or kinetics of spontaneous EPSCs (sEPSCs, Supplementary Fig. 2), suggesting that SKF primarily acted on presynaptic $GABA_B$-receptors.

Deletion of all neurexins from calyx synapses significantly decreased EPSC amplitudes and increased the PPR of evoked EPSCs (Fig. 1c, d), as reported, previously[34]. This decrease is due to a disorganization of active zones after the deletion of neurexins, leading to a loss of presynaptic $Ca^{2+}$-channels, BK-channels, and bassoon[34]. Since $GABA_B$-receptors are localized to active zones[39], we hypothesized that the pan-neurexin deletion may impair the function of presynaptic $GABA_B$-receptors at the presynaptic active zone. Indeed, application of SKF no longer significantly reduced the EPSC amplitude in neurexin-deficient synapses (Fig. 1d, e) or increased the PPR (Fig. 1f). Similar as in control mice, SKF had no effect on the properties of sEPSCs in Nrxn123 TKO mice (Supplementary Fig. 2). Therefore, deletion of neurexins impaired the inhibitory function of presynaptic $GABA_B$-receptors at the calyx of Held synapse.

The major mechanism of action of $GABA_B$-receptors is to inhibit presynaptic $Ca^{2+}$-channels[18,19,38,40]. To test whether the pan-neurexin deletion affects the $GABA_B$-receptor-induced

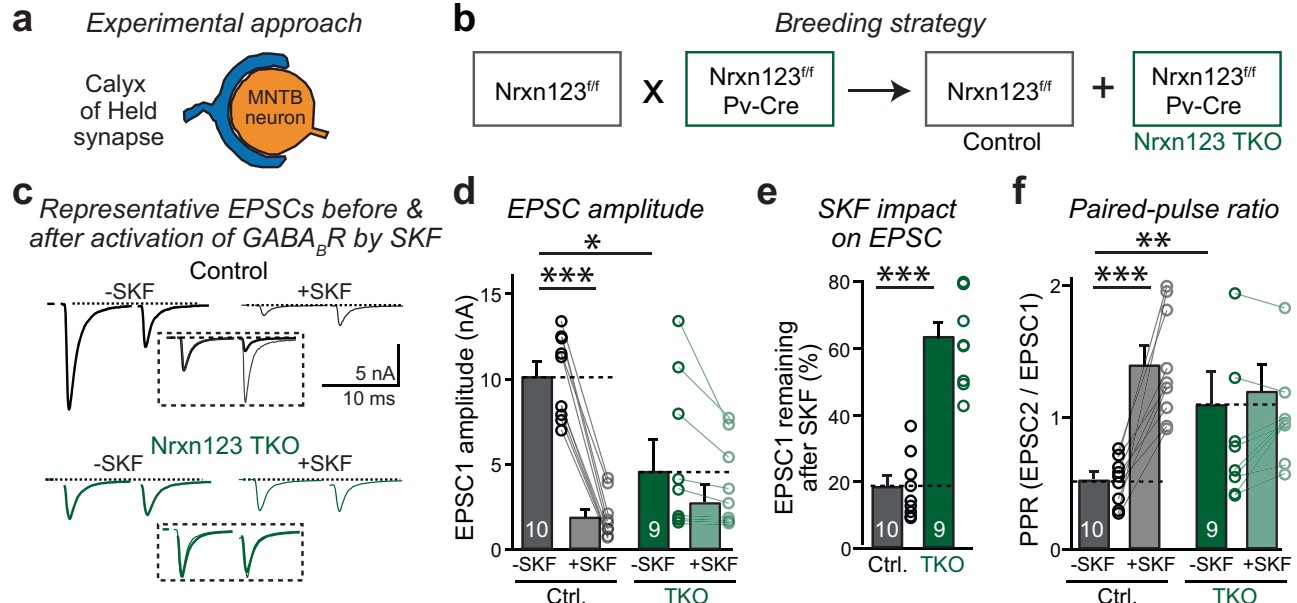

**Fig. 1 Neurexins are required for intact functional organization of GABA$_B$-receptors at the calyx of Held. a** The diagram of the calyx of Held synapse. **b** Strategy for selective deletions of all neurexins at the calyx of Held by crossing PV-Cre mice with triple Nrxn123 cKO mice (Chen et al.[33], Luo et al.[34]). **c** Representative traces of EPSC before and after application of 20 μM SKF-97541 (SKF), a potent and selective GABA$_B$-receptor agonist, recorded in acute slices from littermate control and neurexin123 TKO mice at P12–P14. The normalized EPSCs before and after SKF are shown in inset. **d** Summary graphs of EPSC1 amplitudes before and after SKF for control and Nrxn123 TKO mice. $P = 2.3E{-}7$, $P = 0.072$, paired two-sided $t$-test. $P = 0.0062$, unpaired two-sided $t$-test. **e** Summary graphs of EPSC1 remaining unblocked by SKF application. $P = 1.79E{-}7$, unpaired two-sided $t$-test. **f** Summary graphs of the paired-pulse ratio (PPR) before and after SKF in control and Nrxn123 TKO mice. $P = 2.81E{-}5$, $P = 0.09$, paired two-sided $t$-test. $P = 0.0038$, unpaired two-sided $t$-test. Data are means ± SEM. Number of cells (from at least three mice per group) analyzed are indicated in the bars (**d–f**); Statistical differences were assessed by Student's $t$-test. (*$P < 0.05$; **$P < 0.01$; ***$P < 0.001$). Source data are provided as a Source Data file.

suppression of Ca$^{2+}$-currents, we directly measured depolarization-induced Ca$^{2+}$-currents in patched calyx terminals (Fig. 2). SKF decreased total Ca$^{2+}$-currents by ~38% in control terminals without altering their I–V relationship (Fig. 2). The pan-neurexin deletion had no effect on total Ca$^{2+}$-currents[34], but alleviated the SKF-induced decrease in Ca$^{2+}$-currents, reducing them only by ~19%, again without altering their I–V relationship (Fig. 2). Thus, the pan-neurexin deletion directly counteracts the activity of presynaptic GABA$_B$-receptors on presynaptic Ca$^{2+}$-channels.

To determine whether the decrease in GABA$_B$-receptor function in Nrxn123 TKO synapses is associated with a loss of GABA$_B$-receptor protein from active zones, we analyzed GABA$_B$-receptors by immunocytochemistry. Since GABA$_B$-receptors are known to exist in both presynaptic and postsynaptic compartments as well as in astrocytes, we co-stained the calyx of Held with presynaptic marker VGluT1 and with antibodies against GABA$_{B1}$-receptors or GABA$_{B2}$-receptors. We restricted our analysis only on the presynaptic GABA$_B$-receptors and found a modest but significant decrease in GABA$_{B1}$-receptor and GABA$_{B2}$-receptor proteins in calyx terminals in Nrxn TKO mice, as compared to control (Fig. 3a–d). In contrast, the overall staining of GABA$_B$-receptors was similar in control and Nrxn TKO mice, confirming that deletion of neurexins may only impact on presynaptic GABA$_B$-receptors.

To further examine the spatial distribution of GABA$_B$-receptors at the calyx of Held terminals, we performed high-resolution dSTORM imaging. Because of the transsynaptic nanocolumn alignment of presynaptic active zone and PSD molecules[41], we co-labeled the calyx of Held synapse with Homer 1 and GABA$_{B1}$- or GABA$_{B2}$-receptors. Distribution of GABA$_B$-receptors was quantified and compared in direct opponent to

distribution of Homer 1 (Fig. 4 and Supplementary Fig. 3). No difference was found in the clusters of Homer 1 including their volume and particle count (Supplementary Fig. 3). However, the pan-neurexin knockout resulted in a significant reduction in GABA$_{B1}$ or GABA$_{B2}$ clusters, as compared to control mice (Fig. 4b, e). Interestingly, the volume of GABA$_{B2}$ cluster appeared slightly but significantly larger after pan-neurexin deletion (Supplementary Fig. 3), hinting a looser clustering of GABA$_B$-receptors. Together, these data suggest that neurexins not only impact the function of GABA$_B$-receptors but also their distribution in the calyx terminals.

**The pan-neurexin deletion impairs presynaptic GABA$_B$-receptor functions in excitatory hippocampal Schaffer-collateral synapses.** To test whether the pan-neurexin deletion also affects the function of presynaptic GABA$_B$-receptors at other synapses, we studied excitatory Schaffer-collateral synapses formed by CA3 pyramidal cells on CA1 pyramidal neurons, arguably the best-studied synapse in the brain, which is also known to be regulated by GABA$_B$-receptors[19]. We performed stereotactic injection bilaterally in CA3 regions of Nrxn123 conditional TKO mice at P21 with AAVs encoding ΔCre-eGFP (control) or Cre-eGFP (Nrxn123 TKO). Only mice with successful bilateral AAV injections, as confirmed by monitoring the eGFP expression in the CA3 region, were analyzed by acute slice physiology at P35–P42 (Fig. 5a). In these analyses, we employed extracellular stimulation to induce action potentials in Schaffer collaterals, and recorded EPSCs in CA1 pyramidal neurons using whole-cell patch-clamp recordings (Fig. 5a).

Input/output measurements revealed a modest but significant decrease in synaptic strength in Nrxn123 TKO mice as compared

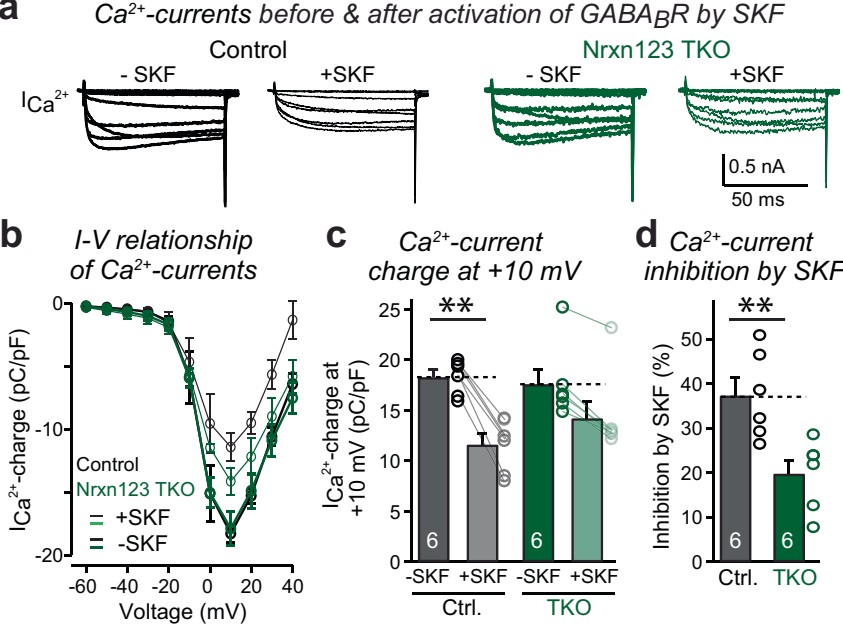

**Fig. 2 Neurexins are required for the modulation of presynaptic Ca$^{2+}$-channels by GABA$_B$-receptors at the calyx terminal. a** Example traces of the presynaptic Ca$^{2+}$ currents induced by a 50 ms step depolarization (from −60 mV to +40 mV in 10 mV increments) before and after application of 10 μM SKF, recorded from the calyx terminals in acute slices from littermate control and Nrxn1/2/3 TKO mice at P12–P14. **b** Summary graphs of the current-voltage relationship of presynaptic Ca$^{2+}$ currents before (thick) and after (thin) activating GBAB$_B$-receptors by SKF. **c** Summary graphs of the current density measured by depolarization to +10 mV before and after SKF. $P = 1.8E-6$, $P = 0.039$, paired two-sided $t$-test. $P = 0.61$, unpaired two-sided $t$-test. **d** Summary graphs of Ca$^{2+}$ current inhibition by SKF. $P = 0.0041$, unpaired two-sided $t$-test. Data are means ± SEM. Number of cells (from at least three mice per group) analyzed are indicated in the bars (**c**, **d**); Statistical differences were assessed by Student's $t$-test. (**P < 0.01). Source data are provided as a Source Data file.

to control mice (Fig. 5b, c), confirming the successful deletion of neurexins. In addition, the rise time of EPSCs was significantly increased by the Nrxn123 KO, suggesting a major underlying change in neurotransmitter release (Fig. 5d, f). The modest effect of the pan-neurexin deletion on synaptic strength in Schaffer-collateral synapses is likely due to the relatively high Ca$^{2+}$-concentration we used following electrophysiological tradition[42], which occludes the loss of Ca$^{2+}$-channels from active zones induced by the pan-neurexin deletion, as described for the pan-neurexin deletion from calyx synapses[34].

Consistent with previous studies[19,30,43], in control mice application of SKF caused a strong depression of EPSC amplitudes (Fig. 5d, e, g), a prominent increase in the PPR (Fig. 5d, h), and an enhancement of the coefficient of variation (CV) of EPSC amplitudes (Fig. 5i). These observations show that activation of GABA$_B$-receptors markedly reduces the release probability at Schaffer-collateral synapses. In Nrxn123 TKO mice, SKF still suppressed EPSCs, but significantly less so than in control mice (Fig. 5d, e, g). Moreover, the Nrxn123 TKO largely prevented the SKF-induced increase in the PPRs (Fig. 5h) and enhancement of the CV of EPSC amplitudes (Fig. 5i), suggesting that deletion of neurexins also impairs the inhibitory function of presynaptic GABA$_B$-receptors at Schaffer-collateral synapses. Similar to calyx synapses, we found that SKF had no effect on sEPSC amplitude or frequency in both control and neurexin-deficient Schaffer-collateral synapses (Supplementary Fig. 4a, 4b).

Together, these results demonstrate that the deletion of all neurexins impairs the function of presynaptic GABA$_B$-receptors at the excitatory synapses.

**The pan-neurexin deletion disrupts the function of presynaptic GABA$_B$-receptors in hippocampal inhibitory synapses.** Because the activation of GABA$_B$-receptors can strongly reduce the release probability of both excitatory synapses and inhibitory synapses[17], we asked whether neurexins are also required for the proper function of GABA$_B$-receptors as autoreceptors in inhibitory synapses. We first examined Pv$^+$ inhibitory synapses in hippocampus, and selectively stimulated PV$^+$ interneurons in hippocampal CA1 with an optogenetic approach (Fig. 6a, top). Using stereotactic injection of AAVs, we expressed Cre-dependent Chief-TdTomato in Pv-Cre mice that are also homozygous for the triple Nrxn123 conditional KO allele (the same mice we used for the experiments on calyx synapses) (Fig. 6a). As a control, we used Pv-Cre mice lacking Nrxn123 conditional KO alleles. Note the control and test mice were not littermates, the Pv-Cre control mice used in these experiments were generated from Pv-Cre/Nrxn123 TKO mice by crossing with wild-type mice of a similar genetic background[33]. We infected mice at P21, and analyzed acute slices derived from these mice at P35-42. In control mice, optogenetic stimulation using 0.5 ms pulses of blue light reliably triggered large IPSCs recorded from CA1 pyramidal cells (Fig. 6b), which are primarily GABAergic (Supplementary Fig. 5). Activation of GABA$_B$-receptors by SKF caused a robust decrease (~45%) of IPSC amplitudes (Fig. 6b–d), a significant increase (~25%) in the PPRs (Fig. 6b, e), and an enhancement (~120%) of the CV of IPSC amplitudes (Fig. 6f). These results confirm that GABA$_B$-receptor activation effectively decreases the release probability at inhibitory synapses under normal conditions. In mice with selective deletion of all neurexins from Pv$^+$ inter-neurons, however, SKF induced a significantly smaller blocking effect (~20%) on IPSCs (Fig. 6b–d). In addition, SKF no longer induced a significant change in either the PPR (Fig. 6e) or the CV of IPSC amplitudes (Fig. 6f). Together, these data suggest that neurexins are required for the normal function of presynaptic GABA$_B$-receptors at the inhibitory synapse.

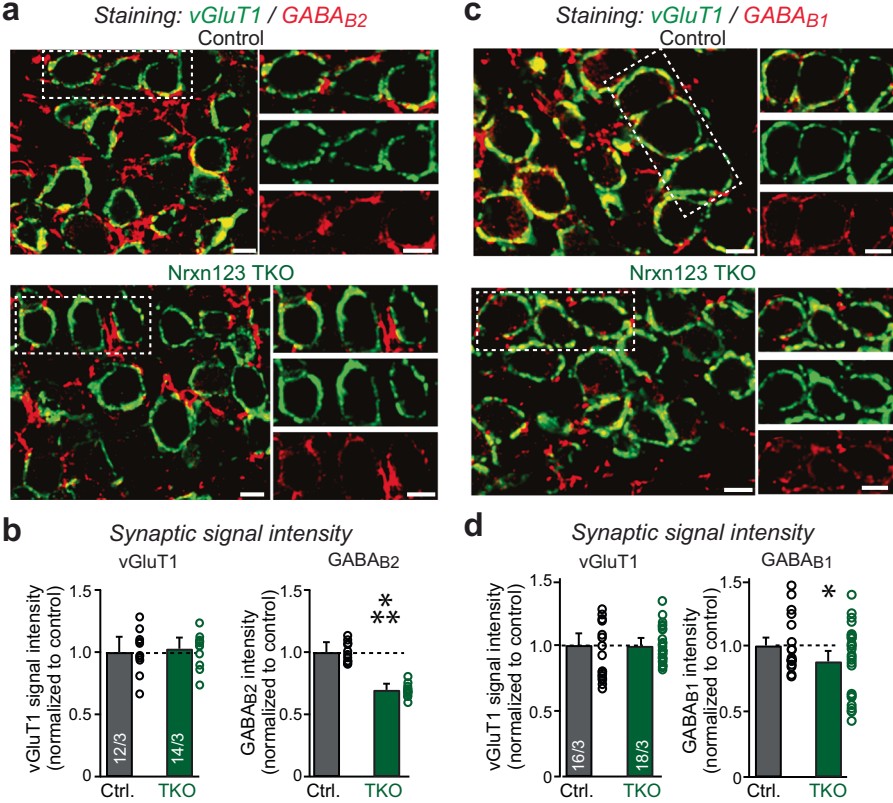

**Fig. 3 Neurexins are required for the expression/localization of GABAB-receptors at the presynaptic nerve terminals. a** Representative confocal microscopy images of MNTB-containing brainstem slice with specific labeling of VGluT1 (green) and GABAB-receptor subunit 2 (GABAB2, red) from both littermate control and Nrxn123 TKO mice at P12. Scale bar, 10 μm. **b** Summary of VGluT1 and GABAB2 immunostaining intensity (normalized to control). $P = 1$, $P = 1.35E−9$, unpaired two-sided $t$-test. **c, d** Same as **a** and **b** except for specific labeling of VGluT1 (green) and GABAB-receptor subunit 1 (GABAB1, red). $P = 1$, $P = 0.031$, unpaired two-sided $t$-test. Data are means ± SEM. Number of sections/animals for immunostaining are indicated in the bars (**b, d**). Statistical differences were assessed by Student's $t$-test. (*$P < 0.05$; ***$P < 0.001$). Source data are provided as a Source Data file.

Surprisingly, application of SKF significantly reduced both the amplitude and frequency of spontaneous IPSCs (sIPSCs) in control mice but not in Nrxn123 TKO mice (Supplementary Fig. 4c, d). Because Pv+ interneurons mostly innervate the pyramidal cell peri-somatically, producing large sIPSCs, the selective loss of Pv+ interneuron-derived sIPSCs may result in the reduction in sIPSC amplitude. However, a postsynaptic effect of SKF could not be excluded. Furthermore, the frequency of sIPSCs also decreased in Nrxn123 TKO mice as compared to control mice in the absence of SKF, confirming the successful deletion of neurexins (Supplementary Fig. 4c, 4d).

**The pan-neurexin deletion impairs the function of presynaptic GABAB-receptors at inhibitory Pv+ basket-cell synapses in the cerebellum.** Finally, to assess whether the role of neurexins in controlling GABAB-receptor function is truly universal, we analyzed inhibitory synapses established by cerebellar basket cells on Purkinje cells (Fig. 7a). Since basket cells are Pv+ interneurons[44,45], we compared conditional triple Nrxn123 KO mice lacking (control) or containing Pv-Cre expression (Nrxn123 TKO). By stimulating basked cell axons with an electrode placed close to the soma of a Purkinje cell, large GABAergic IPSCs can be evoked in an all-or-none manner[46] (Supplementary Fig. 6). As compared to controls, afferent-fiber stimulation-evoked IPSCs were significantly smaller in Nrxn123 TKO mice (Fig. 7b, c), confirming the effective deletion of neurexins at Pv+ basket cells. Interestingly, deletion of neurexins caused a significant reduction in sIPSC frequency but

not sIPSC amplitude (Supplementary Fig. 7). Application of SKF strongly depressed IPSCs in control synapses (Fig. 7c, d) and caused a significant increase in the PPR (Fig. 7e) and the CV of IPSC amplitudes (Fig. 7f). In contrast, deletion of all neurexins alleviated the inhibitory effect of SKF on IPSC amplitudes (Fig. 7d) as well as the SKF-induced increase in the PPR (Fig. 7e) and the CV of IPSC amplitudes (Fig. 7f), suggesting that neurexins are also required for the proper function of GABAB-receptors at cerebellar inhibitory synapses. Similar to the calyx of Held synapse, we found that application of SKF had no effect on the amplitude, frequency, or kinetics of sIPSCs in control or Nrxn123 TKO synapses (Supplementary Fig. 7).

## Discussion
By examining four different central synapses, comprising two excitatory and two inhibitory synapses, we here provide strong evidence that neurexins universally regulate presynaptic GABAB-receptor functions and thereby shape synaptic transmission and synaptic plasticity. Activation of presynaptic GABAB-receptors in all control synapses caused a prominent reduction in release probability, as reflected by a decreased postsynaptic response, increased PPRs, and enhanced CVs of postsynaptic currents (Figs. 1 and 5–7). In neurexin-deficient synapses, activation of presynaptic GABAB-receptors produced a much smaller suppression of release probability, leading to a decreased inhibition of postsynaptic responses, a less pronounced elevation in the PPRs, and the CVs of postsynaptic currents (Figs. 1 and 5–7). The

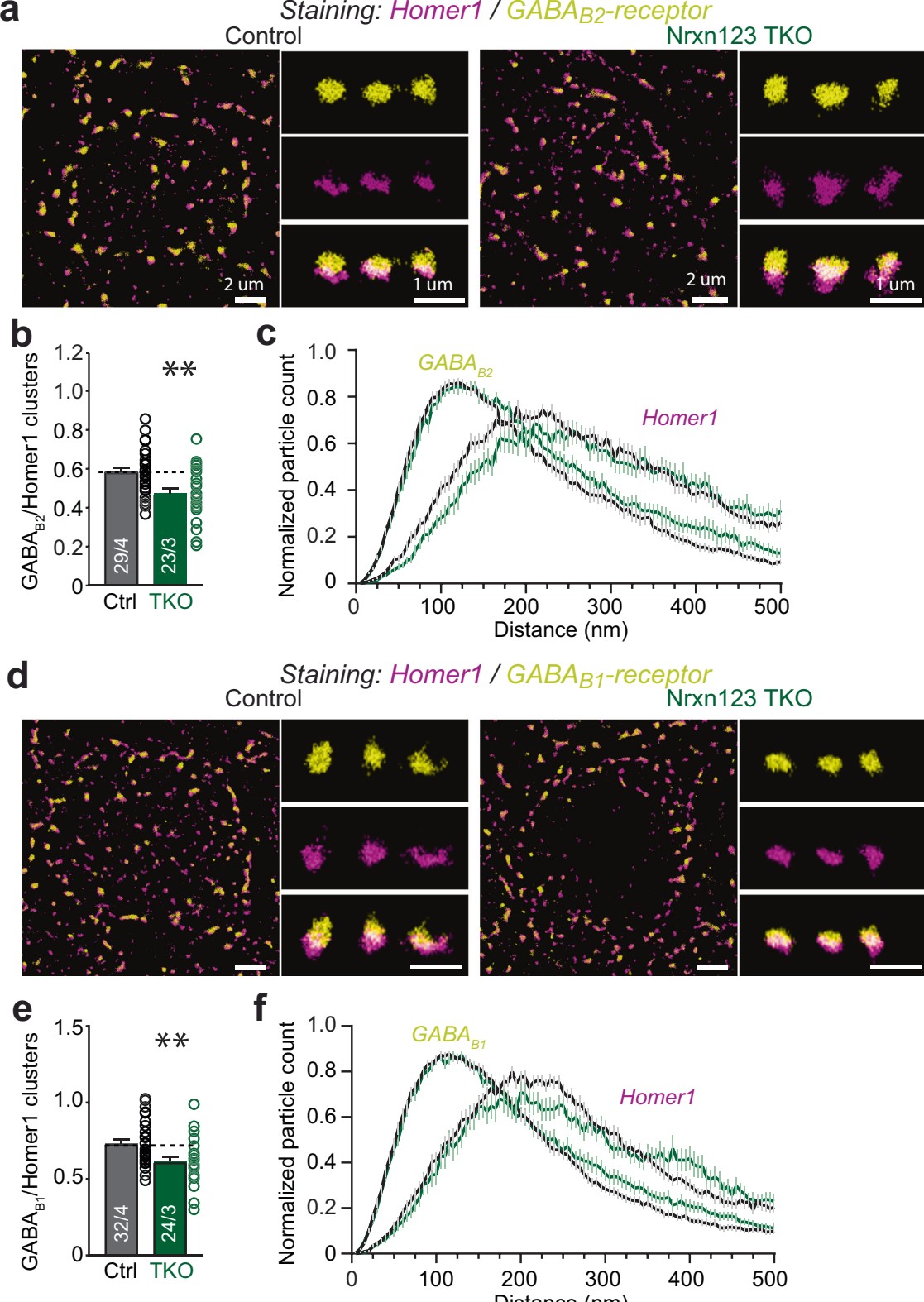

**Fig. 4 Super-resolution dSTORM imaging of GABA_B-receptors at the calyx of Held terminals. a** Representative dSTORM images of MNTB-containing brainstem slice with dual labeling of Homer1 (magenta) and GABA_B-receptor subunit 2 (GABA_{B2}, yellow) from both littermate control and Nrxn123 TKO mice. **b** Summary of the number of GABA_{B2} clusters normalized to the number of Homer1 clusters. $P = 0.0051$, unpaired two-sided $t$-test. **c** Distribution of Homer1 and GABA_{B2} localizations. **d–f** Similar as **a–c**, except for GABA_{B1} antibody used. $P = 0.0057$, unpaired two-sided $t$-test. Data are means ± SEM. Number of sections/animals for immunostaining are indicated in the bars (**b**, **e**). Statistical differences were assessed by Student's $t$-test. (**P < 0.01). Source data are provided as a Source Data file.

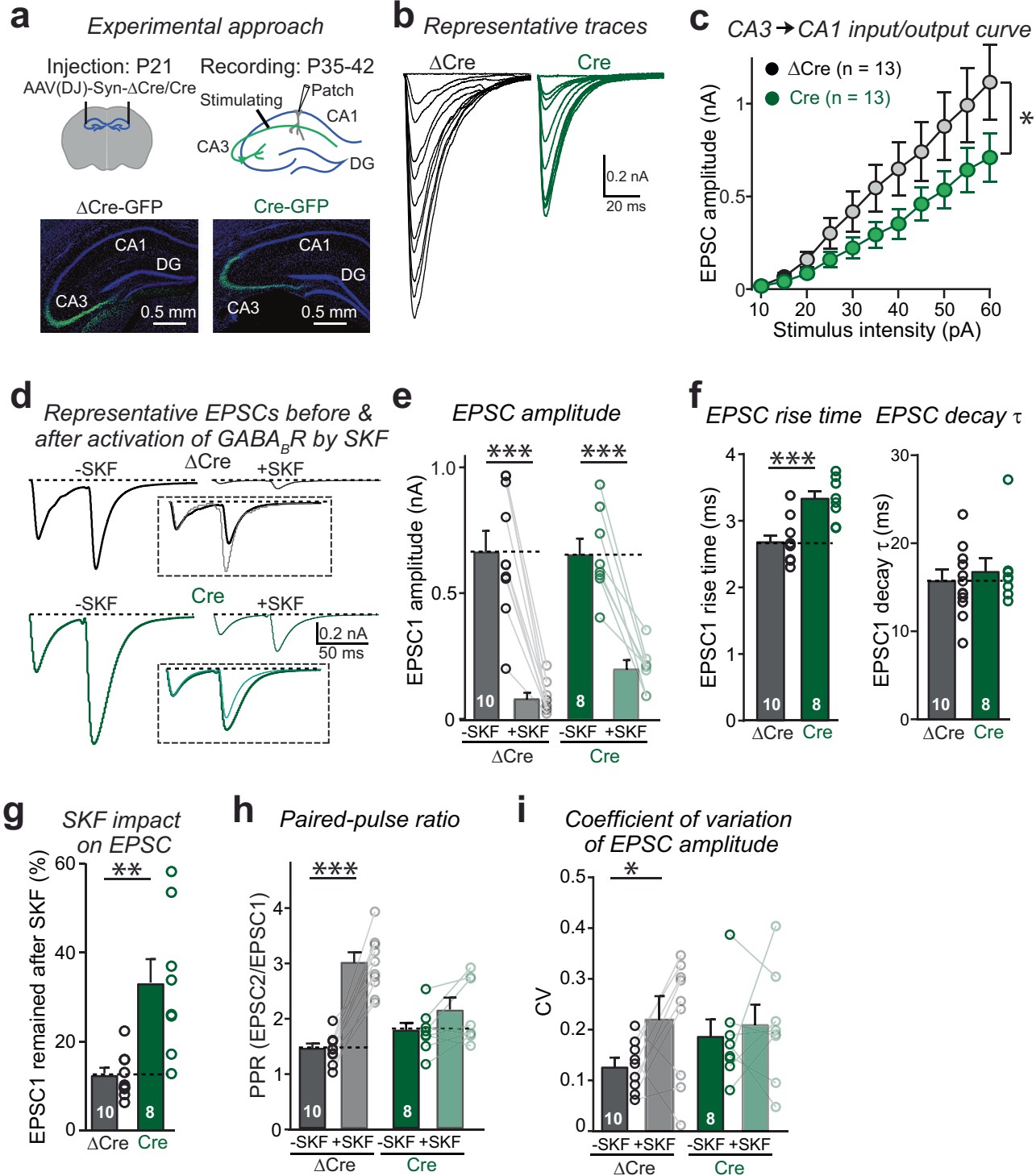

magnitude of the effects differed between synapses in that the cerebellar inhibitory synapses exhibited the largest impairment in GABA$_B$-receptor-mediated responses (Fig. 7) whereas the inhibitory synapses in the hippocampal CA1 region had the least impressive phenotype (Fig. 6), probably due to large heterogeneity of distinct GABAergic synapses[33,47]. However, even in Pv$^+$ inhibitory synapses onto CA1 pyramidal cells, the deletion of neurexins completely abolished the strong increase in the PPR and CV of IPSCs induced by GABA$_B$-receptor activation, suggesting an impairment in the function of GABA$_B$-receptors at these synapses (Fig. 6).

Accumulating evidence demonstrates that presynaptic GABA$_B$-receptors are expressed in almost all excitatory and inhibitory synapses. GABA$_B$-receptors are primarily coupled to $G_{i/o}$ proteins. Activation of presynaptic GABA$_B$-receptors either by specific agonists or by endogenous GABA profoundly inhibits synaptic transmission in most synapses studied[9,15,16,18,21,48,49], but see ref. [47]. One major mechanism by which presynaptic GABA$_B$-receptors regulate release probability is mediated by $G_{\beta\gamma}$ direct binding to and inhibiting Ca$^{2+}$-channels[50]. Consistent with the primary effect of GABA$_B$-receptors in inhibiting presynaptic Ca$^{2+}$-channels[15,18,19,38], our direct recordings of Ca$^{2+}$-currents

**Fig. 5 Deletion of neurexins impairs the function of GABA$_B$-receptors at CA3-CA1 excitatory synapses. a** Schematic of experimental approach for virus injection and electro-physiology recording (Top); GFP expression in AAV-transfected CA3 region of hippocampus (bottom). **b** Representative traces of EPSCs evoked by fiber stimulations with increased intensity, recorded in acute hippocampus slices from Nrxn123 cKO mice injected with AAV-ΔCre or AAV-Cre. **c** The input–output curve of EPSC amplitudes in relation to the stimulation intensity. $P = 0.0113$, unpaired two-sided $t$-test. **d** Example traces of paired-pulse EPSCs before and after addition of GABA$_B$-receptor agonist SKF, the normalized EPSCs before and after SKF are shown in inset. The intensity of fiber stimulation was tuned to evoke EPSC1 at similar amplitude for each cell. **e** Summary graphs of EPSC1 amplitudes before and after SKF for control (ΔCre) and Nrxn123 TKO synapses (Cre). $P = 0.0001$, $P = 0.0001$, paired two-sided $t$-test. $P = 0.99$, unpaired two-sided $t$-test. **f** Summary graphs of EPSC1 rise time (left) and decay time constants (right). $P = 0.0008$, $P = 0.6136$, unpaired two-sided $t$-test. **g** Summary graphs of EPSC1 remaining unblocked by SKF application. $P = 0.0013$, unpaired two-sided $t$-test. **h** Summary graphs of the PPR before and after SKF in control and Nrxn123 TKO synapses. $P = 0.0001$, $P = 0.0984$, paired two-sided $t$-test. $P = 0.2772$, unpaired two-sided $t$-test. **i** Summary graphs of the CV of EPSC1 amplitude before and after SKF in control and Nrxn123 TKO synapses. $P = 0.0385$, $P = 0.99$, paired two-sided $t$-test. $P = 0.4664$, unpaired two-sided $t$-test. Data are means ± SEM. Number of cells (from at least three mice per group) analyzed are indicated in the graph (**c**) or bars (**e–i**); Statistical differences were assessed by Student's $t$-test. (*$P < 0.05$; **$P < 0.01$, ***$P < 0.001$). Source data are provided as a Source Data file.

in calyx terminals showed that activation of GABA$_B$-receptors by SKF inhibited Ca$^{2+}$ influx in control synapse. Such SKF-induced inhibition of Ca$^{2+}$ currents was significantly smaller in neurexin-deficient calyx synapses (Fig. 2). Thus, although the neurexin deletion did not change by itself the magnitude of presynaptic Ca$^{2+}$-currents[34], it altered the sensitivity of Ca$^{2+}$-currents to GABA$_B$-receptor activation.

Strong studies have demonstrated the essential function of neurexins in shaping diverse synaptic properties in various animal species and preparations, although uncertainty still reigns about the precise role and mechanism of different isoforms of neurexins[32]. It has been shown that deletions of all neurexins in different synapses produce severe but dramatically different synaptic phenotypes[33,34], ranging from reduced synapse numbers, decreased Ca$^{2+}$-influx during action potentials, to decoupling of Ca$^{2+}$-channels with synaptic vesicles. It seems that different neurexins may act as essential synaptic organizers in modulating different aspects of synaptic properties[32,33]. Such modulatory functions could be diverse and synapse-specific, depending on the unique isoforms and alternative splicing of neurexins as well as the presence of their interacting partners, which may explain the observed large variability of pan-neurexin deletion induced synaptic phenotypes in our current work and previous studies[33,34]. The consistent role of GABA$_B$-receptors in synaptic transmission and the crucial requirement for neurexins in this role of GABA$_B$-receptors that we have observed in all four synapses suggest a universal function of neurexins in organizing active zone signaling complexes. In the future, systematic studies of the function of specific neurexins at different synapses are necessary to address how all these mechanisms are integrated to regulate synaptic transmission and synaptic plasticity.

Another important question is how neurexins mediate the proper signaling of presynaptic GABA$_B$-receptors on the function of Ca$^{2+}$ channels at the active zone. Two potential mechanisms may underlie the effect of pan-neurexin deletion on GABA$_B$-receptor inhibition of Ca$^{2+}$ channels and thus neurotransmitter release. First, mis-localization of Ca$^{2+}$ channels in neurexin-deficient synapse[34] may simply increase their distance to GABA$_B$-receptor cluster and therefore render the inhibition less sensitive. However, we think that this possibility is highly unlikely because pan-neurexin deletion causes no change in Ca$^{2+}$ channels, including their number and distribution, in mPFC PV$^+$ interneurons[33]. Second, pan-neurexin deletion may lead to a direct impairment in the distribution or functioning of GABA$_B$-receptors. Our immunohistochemistry analysis (Fig. 3) and dSTORM imaging of GABA$_B$-receptors (Fig. 4) at the calyx of Held terminals reveal a significant reduction in the abundance and localization of presynaptic GABA$_B$-receptors, suggesting that neurexins play an important role in the distribution of GABA$_B$-

receptors. Moreover, we observe a strong ablation of GABA$_B$-receptor inhibition on Ca$^{2+}$ channels in both hippocampal and cerebellar PV$^+$ interneurons (Figs. 6 and 7). The effects on the abundance and localization of presynaptic GABA$_B$-receptors appear to be modest, as compared to the strong impact on inhibition of EPSC/IPSC, which thus may warrant an additive role of neurexin in modulating the functioning of GABA$_B$-receptors.

Since direct interaction between neurexins and GABA$_B$-receptors is lacking, it is likely that neurexins may bind with other molecules, intracellularly and/or extracellularly, to facilitate the localization, stabilize the membrane expression, or enable the function of GABA$_B$-receptors. Further studies are required to address these issues by identifying the interacting proteins with which neurexins specifically organize the localization and function of GABA$_B$-receptors in the presynaptic active zone. It is expected that a complex molecular network may be involved. Several studies have been conducted to identify the interactome of GABA$_B$-receptors. Native GABA$_B$-receptors form macromolecular complexes containing multiple interacting proteins, including Ca$^{2+}$-channels, AJAP1, APP, and calsyntenin[13]. These proteins are thought to regulate GABA$_B$-receptor trafficking, expression and signaling[13,26]. APP and secreted APP$_\alpha$ bind to the sushi domain of GABA$_{B1a}$[29,30]. Deletion of APP may cause a reduction in the expression of GABA$_B$-receptors and a consequent alleviation of the inhibition of transmitter release by GABA$_B$-receptors[29,30]. Similarly, deletion of FMR1 produces a selective decrease in the expression level of GABA$_{B1a}$ and a significant impairment in GABA$_B$-receptor-dependent presynaptic inhibition of neurotransmitter release in hippocampal excitatory synapses[43]. Such an impact of FMR1 on the presynaptic GABA$_B$-receptor signaling was not found in hippocampal inhibitory synapses[43]. Interestingly, calsyntenin-3, one of the integrated GABA$_B$-receptor macromolecule complexes[26], has been shown to interact with both α-neurexin and β-neurexin extracellularly[51,52] and to mediate GABAergic and glutamatergic synapse formation[52]. Neurexophilin, another α-neurexin ligand with restricted expression in subpopulations of inhibitory neurons, has also been reported to support presynaptic GABA$_B$-receptor function[53]. Our findings thus strengthen the hypothesis that neurexins are central active zone organizers that orchestrate presynaptic signaling networks, including Ca$^{2+}$-channel complexes and GABA$_B$-receptor complexes. On the other hand, neurexins are known to interact with diverse trans-synaptic ligands to shape many postsynaptic signaling complexes, such as AMPA-receptors, NMDA-receptors, and endocannabinoid signaling[54–56]. Therefore, a coherent picture is emerging that neurexins may integrate presynaptic and postsynaptic signaling complexes for specific and precise regulation of synaptic properties[32,57].

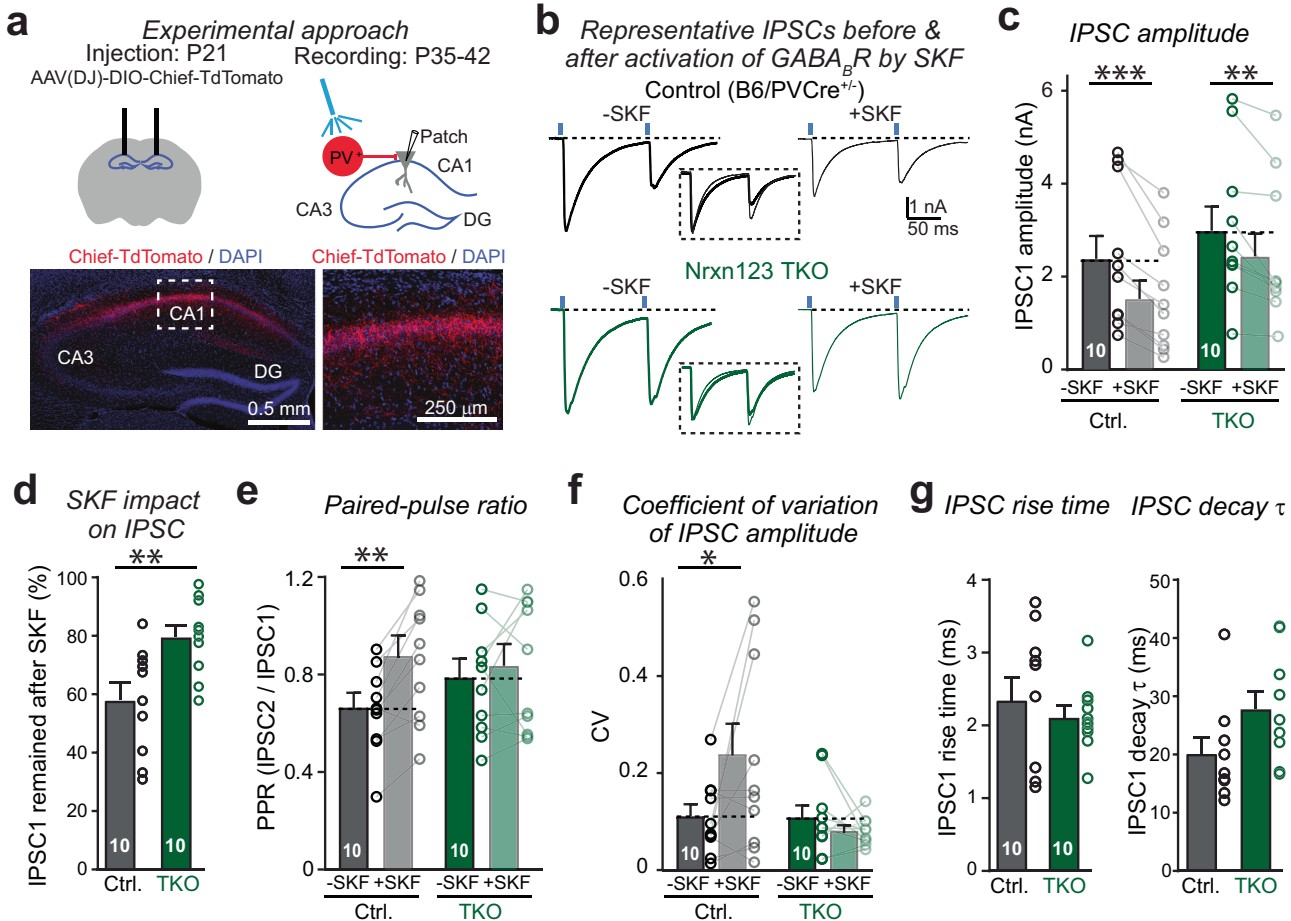

**Fig. 6 Deletion of neurexins impairs the function of GABA_B-receptors at the inhibitory synapse between PV interneurons and CA1 pyramidal cells in the hippocampus. a** Schematic of experimental setup for virus injection and optogenetic stimulation of PV$^+$ interneurons (Top); PV-Cre dependent expression of Chief-TdTomato in AAV-transfected CA1 region of hippocampus (bottom). **b** Example traces of paired-pulse blue light-evoked IPSCs before and after addition of GABA_B-receptor agonist SKF, recorded in acute hippocampus slices from PVCre$^+$ mice (control) or PVCre$^+$/Nrxn123 cKO mice (TKO) injected with AAV-DIO-Chief-TdTomato. The normalized IPSCs before and after SKF are shown in inset. **c** Summary graphs of IPSC1 amplitudes before and after SKF for control and Nrxn123 TKO synapses. $P = 0.0001$, $P = 0.0026$, paired two-sided $t$-test. $P = 0.6$, unpaired two-sided $t$-test. **d** Summary graphs of IPSC1 amplitude remaining unblocked by SKF. $P = 0.0047$, unpaired two-sided $t$-test. **e** Summary graphs of the PPR ratio before and after SKF in control and Nrxn123 TKO synapses. $P = 0.0022$, $P = 0.6165$, paired two-sided $t$-test. $P = 0.5114$, unpaired two-sided $t$-test. **f** Summary graphs of the CV of IPSC1 amplitude. $P = 0.0143$, $P = 0.99$, paired two-sided $t$-test. $P = 0.99$, unpaired two-sided $t$-test. **g** Summary graphs of IPSC1 rise time (left) and decay time constant (right). $P = 0.4974$, $P = 0.063$, unpaired two-sided $t$-test. Data are means ± SEM. Number of cells (from at least three mice per group) analyzed are indicated in the bars (**c–g**); Statistical differences were assessed by Student's $t$-test. (*$P < 0.05$; **$P < 0.01$; ***$P < 0.001$). Source data are provided as a Source Data file.

## Methods

**Mouse breeding, genotyping, and husbandry**. All experiments were approved by the Institutional Animal Care and Use Committee at Stanford University. All experiments were performed using littermates of either sex. Triple Nrxn123 conditional KO mice were crossed with PV-IRES-Cre driver line to generate cell-specific Nrxn123 deletion at the calyx of Held synapse[33,34], hippocampal PV$^+$ interneurons and cerebellar basket cells. Mice were housed at room temperature and 40–60% humidity on a 12 h light–dark cycle (7:00 to 19:00, light) with food and water freely available. No statistical tests were used to predetermine sample size because the effect size was not known before the experiments, and determining the effect size would have required as many mice as the actual experiments. All experiments were performed blindly without knowledge by the experimenter of the mouse genotypes. All experiments on the calyx of Held were performed using P12–14 mice while those on hippocampal and cerebellar neurons using P35–42 mice.

The primer sequences used for genotyping were:
Nrxn1 flox: 5′ GTAGCCTGTTTACTGCAGTTCATTCC 3′ and
5′ CAAGCACAGGATGTAATGGCCTTTC 3′
Nrxn2 flox: 5′ CAGGGTAGGGTGTGGAATGAGGTC 3′ and
5′ GTTGAGCCTCACATCCCATTTGTCT 3′
Nrxn3 flox: 5′ AATAGCAGAGGGGTGTGACAC 3′ and

5′ CGTGGGGTATTTACGGATGAG 3′
Cre: 5′ GAACCTGATGGACATGTTCAGG 3′ and
5′ AGTGCGTTCGAACGCTAGAGCCTGT 3′

**Preparation of brain slices for the calyx of Held electrophysiology**. Coronal brain slices containing the MNTB nucleus were prepared as described previously[58]. In brief, mice of postnatal day 12–14 were decapitated; brains were rapidly isolated and glued on the cutting chamber of a vibratome (VT1200s; Leica), which was immersed in oxygenated cold ACSF containing (in mM): 119 NaCl, 26 NaHCO$_3$, 10 glucose, 1.25 NaH$_2$PO$_4$, 2.5 KCl, 0.05 CaCl$_2$, 3 MgCl$_2$, 2 Na-pyruvate, and 0.5 ascorbic acid, pH 7.4. Transverse 160–200 μm slices were sectioned and transferred into a beaker with bubbled ACSF containing (in mM): 119 NaCl, 26 NaHCO$_3$, 10 glucose, 1.25 NaH$_2$PO$_4$, 2.5 KCl, 2 CaCl$_2$, 1 MgCl$_2$, 2 Na-pyruvate, and 0.5 ascorbic acid, pH 7.4. After recovery at 35 °C for 45 min, slices were stored at room temperature (~21–23 °C) for experiments.

Whole-cell voltage-clamp recordings were made from cells visualized by infrared differential interference contrast (IR-DIC) video microscopy (Axioskop 2; Zeiss). Patch-clamp recording were made with the EPC 10 amplifier (HEKA, Lambrecht, Germany) and the software PatchMaster (HEKA, Lambrecht, Germany). Patch pipettes (resistance of 3–4 MΩ) were pulled using borosilicate glass (WPI) on a two-stage vertical puller (Narishige).

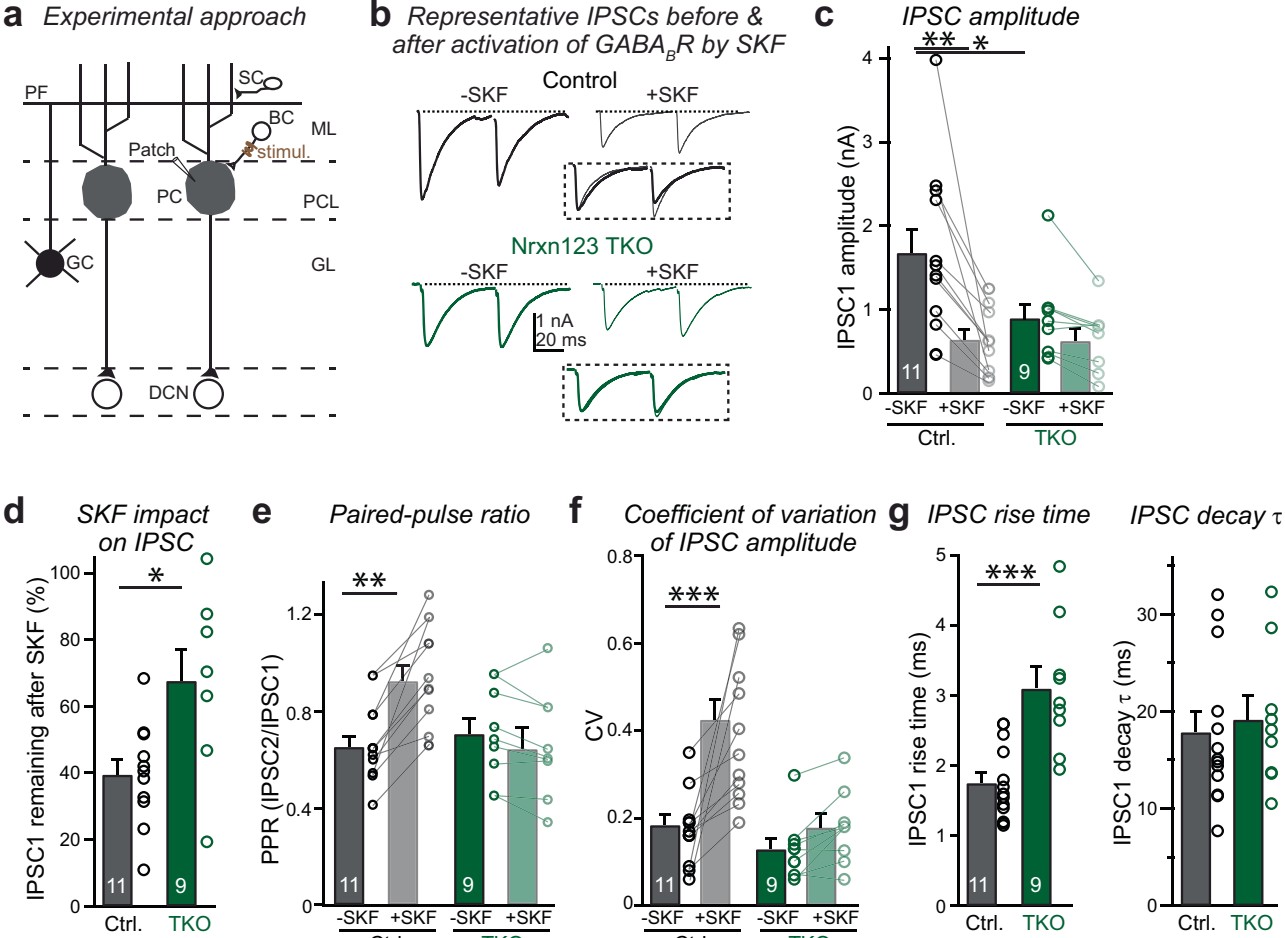

**Fig. 7 Deletion of neurexins impairs the function of GABA_B-receptors at the inhibitory synapse between basket cells and Purkinje cells in the cerebellum. a** Schematic of cerebellum circuits and experimental approach for stimulating basket cell (BC)-Purkinje cell (PC) inhibitory synapse. **b** Example traces of paired-pulse IPSCs before and after addition of GABA_B-receptor agonist SKF, recorded in acute cerebellar slices from littermate control and Nrxn123 TKO mice P35-49. The normalized IPSCs before and after SKF are shown in inset. **c** Summary graphs of IPSC1 amplitudes before and after SKF for control and Nrxn123 TKO synapses. $P = 0.0016$, $P = 0.051$, paired two-sided t-test. $P = 0.0389$, unpaired two-sided t-test. **d** Summary graphs of IPSC1 amplitude remaining unblocked by SKF for control and Nrxn123 TKO mice. $P = 0.0028$; unpaired two-sided t-test. **e** Summary graphs of the PPR before and after SKF in control and Nrxn123 TKO mice. $P = 0.000101$, $P = 0.097$, paired two-sided t-test. $P = 0.762$, unpaired two-sided t-test. **f** Summary graphs of the CV of IPSC1 amplitude before and after SKF in control and Nrxn123 TKO mice. $P = 0.00073$, $P = 0.066$, paired two-sided t-test. $P = 0.348$, unpaired two-sided t-test. **g** Summary graphs of IPSC1 rise time (left) and decay time constant (right). $P = 0.000953$, $P = 0.977$; unpaired two-sided t-test. Data are means ± SEM. Number of cells (from at least three mice per group) analyzed are indicated in the bars (**c–g**); Statistical differences were assessed by Student's t-test. (*$P < 0.05$; **$P < 0.01$; ***$P < 0.001$). Source data are provided as a Source Data file.

For EPSCs recordings, the MNTB cells were voltage-clamped at −70 mV, and EPSCs were recorded in ACSF. Picrotoxin (100 μM), strychnine (2 μM), and D-AP5 (50 μM) were routinely added to block GABA_A-receptors, glycine-receptors, and NMDA-receptors, respectively. The pipette internal solution contained (in mM): 120 Cs-gluconate, 20 tetraethylammonium-Cl, 20 HEPES, 2 EGTA, 4 MgATP, 0.4 NaGTP, 10 phosphocreatine, and 2 Qx-314. Afferent fiber stimulation (0.1–0.5 mA, 0.1 ms) were delivered using a bipolar electrode positioned halfway between midline and the MNTB to ensure that the calyx was activated in an all-or-none manner. Series resistances (<12 MΩ) were compensated by 70–90% in order to maintain a residual resistance of <2 MΩ.

For presynaptic Ca²⁺ current recording, the calyces were patched and voltage-clamped routinely at resting −80 mV in whole-cell mode[59]. Ca²⁺ influx was evoked by a 50 ms step depolarization ranging from −60 mV to + 40 mV. Ca²⁺ currents were isolated pharmacologically with a bath solution containing (in mM): 105 NaCl, 20 TEA-Cl, 2.5 KCl, 1 MgCl₂, 2 CaCl₂, 25 NaHCO₃, 1.25 NaH₂PO₄, 25 glucose, 0.4 ascorbic acid, 3 myo-inositol, 2 Na-pyruvate, 0.001 tetrodotoxin (TTX), 300–310 mOsm, pH 7.4 when bubbled with 95% O₂, and 5% CO₂. The standard pipette solution contained (in mM): 125 Cs-gluconate, 20 CsCl, 4 MgATP, 10 Na₂-phosphocreatine, 0.3 NaGTP, 10 HEPES, 0.05 BAPTA, 310–320 mOsm, pH 7.2 adjusted with CsOH. Patch pipettes with resistance of 3–4 MΩ were used and series resistances, typically less than 20 MΩ, were compensated by 70–90%.

SKF 97541 (Tocris, Cat#: 0379) was prepared as 100 mM stock in distilled water and stored at −20 °C. During experiment, the stock was freshly resolved into ACSF at a final concentration of 20 μM. After 2–3 min baseline recording, the SKF-containing ACSF was perfused in and normally reached maximum effects within 5 min (Supplementary Fig. 1a). The blocking impact of SKF was calculated by comparing the steady-state current amplitude in SKF to the baseline current amplitude of the same neuron.

**Stereotactic injections of AAV in hippocampal CA3.** To delete neurexins in CA3 hippocampal pyramidal cells, we injected AAV-DJ-GFP strain expressing active or inactive cre-recombinase (Cre or ΔCre) under the control of the synapsin promoter[60]. To prepare the virus, AAV plasmids were co-transfected with pHelper and pRC-DJ into HEK293T cells. Seventy-two hours of post-transfection, cells were harvested, lysed and run on an iodixanol gradient by ultracentrifugation at 400,000×g for 2 h. The 40% iodixanol fraction containing AAV was collected, concentrated and washed in a 100 K MWCO ultracon filter.

For stereotactic injections into CA3, conditional Nrxn1/2/3 knockout mice at P21 were anesthetized. AAV expressing GFP-Cre or GFP-ΔCre recombinase was injected with a glass pipette bilaterally into the CA3 region of the hippocampus. Two injection sites per hemisphere were performed, using the following coordinates from the Bregma: AP: −2.1/−2.1 mm, ML: ±2.0/±2.8 mm, DV: −2.2/

−2.1 (flow rate = 0.35 μl/min; injected volume = 0.8 μl). Recordings were performed 2 weeks after injection, around P35-42.

**Stereotactic injections of AAV in hippocampal CA1**. To selectively express Cre-dependent Chief-TdTomato in PV$^+$ positive interneurons in hippocampal CA1, we injected AAV-DJ-DIO-Chief-TdTomato into Nrxn123 cTKO mice lacking or expressing PVCre. The virus was prepared as previously described (see stereotactic injections of AAV in hippocampal CA3). Mice were injected at P21. One injection site per hemisphere was performed, using the following coordinates from the Bregma: AP: −1.8 mm, ML: ±1.45 mm, DV: −1.35 mm (flow rate = 0.35 μl/min; injected volume = 0.8 μl). Recordings were performed 2 weeks after injections, around P35-42.

**Electrophysiology in hippocampal slices**. For acute hippocampal slice electrophysiology, coronal hippocampal sections (250 μm) were cut in ice-cold solution containing (in mM): 228 Sucrose, 2.5 KCl, 1 NaH$_2$PO$_4$, 26 NaHCO$_3$, 0.5 CaCl$_2$, 7 MgSO$_4$-7H$_2$O, 11 D-Glucose saturated with 95% O$_2$/5% CO$_2$. Slices were transferred to a holding chamber containing ACSF (in mM): 119 NaCl, 26 NaHCO$_3$, 11 D-Glucose, 2.5 KCl, 1 NaH$_2$PO$_4$, 2.5 CaCl$_2$, 1.3 MgSO$_4$-7H$_2$O, 290 mOsm. Slices were allowed to recover at 32 °C for 30 min then at room temperature (~21–23 °C) for 1 h. Slices were transferred to a recording chamber perfused with oxygenated ACSF (1.5 ml/min) maintained at 32 °C. Whole-cell voltage-clamp recordings of CA1 pyramidal neurons were performed with the Axon Multiclamp 700 B amplifier and the software Clampex 10.4 (Molecular Devices, USA). Patch pipettes (resistance of 3–3.5 MΩ) were pulled using borosilicate glass (WPI) on a two-stage vertical puller (Narishige). The internal solution contained (in mM): 135 Cesium-Methanesulfonate, 8 NaCl, 10 HEPES, 0.3 EGTA, 2 Mg-ATP, 0.3 Na2-GTP, 7 Phosphocreatine, 0.1 Spermine, 2 QX314, 300 mOsm, pH 7.3 with CsOH.

CA1 pyramidal neurons were voltage-clamped at −70 mV and EPSCs were evoked by stimulating Schaffer collateral fibers with a bipolar electrode. For paired-pulse ratios, EPSCs were evoked with paired pulse stimulations at an inter-stimulus interval of 50 ms. Picrotoxin (100 μM) was added to the bath to inhibit GABA$_A$-receptors. To study the impact of GABA$_B$-receptors on CA3-CA1 synaptic transmission, baseline EPSCs were recorded at −70 mV and the stimulus intensity was adjusted for each cell to evoke a comparable EPSCs of ~500 pA. After 5 min recording baseline EPSCs, 20 μM SKF was washed in to activate GABA$_B$-receptors, while Schaffer collaterals were continuously stimulated.

For recordings of PV interneuron synapses, CA1 pyramidal neurons were voltage-clamped at −70 mV and IPSCs were evoked by selectively triggering action potential firing in PV positive interneurons infected with Cre-dependent Chief using 0.5 ms pulses of blue light. Recordings were performed in the presence of CNQX and AP-5 to inhibit AMPA-receptor and NMDA-receptor respectively.

**Electrophysiology in cerebellar slices**. For acute cerebellar slice recording, parasagittal cerebellar sections of the vermis (250 μm) were cut in ice-cold oxygenated low Ca$^{2+}$ ACSF containing (in mM): 119 NaCl, 26 NaHCO$_3$, 10 glucose, 1.25 NaH$_2$PO$_4$, 2.5 KCl, 0.05 CaCl$_2$, 3 MgCl$_2$, 2 Na-pyruvate, and 0.5 ascorbic acid, pH 7.4. Slices were incubated at room temperature with bubbled ACSF containing (in mM): 119 NaCl, 26 NaHCO$_3$, 10 glucose, 1.25 NaH$_2$PO$_4$, 2.5 KCl, 2 CaCl$_2$, 1 MgCl$_2$, 2 Na-pyruvate, and 0.5 ascorbic acid, pH 7.4. After recovery for at least 1 h, slices were transferred to a recording chamber for experiments. Whole-cell voltage-clamp recordings of Purkinje cells in cerebellar lobules IV/V were performed with HEKA EPC 10 amplifier (HEKA, Lambrecht, Germany) and the software Patch-Master (HEKA, Lambrecht, Germany). Patch pipettes (resistance of 3–4 MΩ) were filled with the internal solution contained (in mM): 120 CsCl, 20 tetra-ethylammonium-Cl, 20 HEPES, 2 EGTA, 4 MgATP, 0.4 NaGTP, 10 phospho-creatine, and 2 Qx-314. Afferent fiber stimulation (0.1–0.5 mA, 0.1 ms) were delivered using a bipolar electrode positioned in the inner molecular layer closed to the soma of Purkinje cells (within 100 μm). Only recordings displaying in all-or-none manner were accepted for further analysis (Zhang et al.[46]). Series resistances (<20 MΩ) were compensated by 70–90% in order to maintain a residual resistance of <2 MΩ. CNQX and AP-5 were added in ACSF to block AMPA-receptor and NMDA-receptor, respectively.

**Quantifications and statistical analyses**. Electrophysiological data were analyzed in Igor Pro (Wavemetrics). For clarity, all stimulus artifacts were blanked and not shown in the figures. All data were shown as means ± SEM. The numbers of analyzed cells from at least two mice per group were shown in the graph as indicated in the figures. Student's $t$-test was used for two-group comparisons. Statistical significance was defined and indicated in the figures and figure legends as follows: $*p < 0.05$; $**p < 0.01$; $***p < 0.001$.

**Immunohistochemistry**. Mice of postnatal day 12–14 were anesthetized and perfused with 1× PBS for 5 min followed by 2–4% paraformaldehyde (PFA) or 5 min. The brains were carefully extracted and post-fixed in 4% PFA for 2 h, followed by immersion in 20–30% sucrose for 48 h for complete cryo-protection. Transverse brain sections at 20–30 μm were cut at −20 °C using a cryostat (CM3050S, Leica). The slices containing the MNTB nucleus were pretreated in 0.5% Triton X-100 and

5% goat serum in PBS for 1 h at room temperature and incubated overnight at 4 °C with primary antibodies in blocking solution (0.1% Triton X-100 and 5% goat serum in PBS). The slices were washed with PBS and incubated with fluorescence-conjugated secondary antibodies for 2 h at room temperature. After wash, the slices were mounted with DAPI fluoromount (SouthernBiotech). Primary antibodies against VGluT1 (guinea pig, polyclonal, 1:1000, Millipore, Cat#: AB5905; RRID: AB_2301751), GABA$_{B1}$ (mouse, monoclonal, 1:500, NeuroMab Cat#: 75-183), and GABA$_{B2}$ (mouse, monoclonal, 1:500, NeuroMab Cat#: 75-125) were used. Secondary antibodies were Alexa Fluor conjugates (1:500; Invitrogen). Images were acquired using Nikon A1RSi confocal microscope with a 60× oil-immersion objective (1.45 numerical aperture) and analyzed in Nikon Analysis software.

**Direct stochastic optical reconstruction microscopy (dSTORM) imaging**. dSTORM images were recorded with a Vutara SR 352 (Bruker Nanosurfaces, Inc., Madison, WI) commercial microscope based on single molecule localization biplane technology[61,62]. Twenty micrometer thick brainstem slices containing the MNTB region were prepared as described and labeled with Homer1 (rabbit, 1:1000, Millipore, Cat#: ABN37), GABA$_{B1}$ (mouse, monoclonal, 1:1000, NeuroMab Cat#: 75-183), GABA$_{B2}$ (mouse, monoclonal, 1:1000, NeuroMab Cat#: 75-125) primary antibodies and secondary antibodies conjugated to Alexa647 (1:3000, Thermo-Fisher) or CF568 (1:3000, Biotium). The slices were mounted on a coverslip coated with poly-L-Lysine and placed in dSTORM buffer containing (in mM) 50 Tris-HCl at pH 8.0, 10 NaCl, 20 MEA, 1% β-mercaptoethanol, 10% glucose, 150 AU glucose oxidase type VII (Sigma Cat#: G2133), and 1500 AU catalase (Sigma Cat#: C40). Labeled proteins were imaged with 647 and 561 nm excitation power of 40 kW/cm². Images were recorded using a 60×/1.2 NA Olympus water immersion objective and Hamamatsu Flash4 sCMOS camera with gain set at 50 and frame rate at 50 Hz. Data was analyzed by Vutara SRX software (version 6.04). Single molecules were identified in each frame by their brightness after removing the background. Identified molecules were localized in three dimensions by fitting the raw data in a 12 × 12-pixel region of interest centered around each particle in each plane with a 3D model function that was obtained from recorded datasets of fluorescent beads. Fit results were filtered by a density based denoising algorithm to remove isolated particles and rendered as 50 nm points. The remaining localizations were classified into clusters by density-based spatial clustering of applications with noise (DBSCAN), a minimum of 30 localizations were connected around a 100 nm search radius. Localizations were rendered as 50 nm points for analysis by Pearson's correlation. The experimentally achieved image resolution of 40 nm laterally ($x$, $y$) and 70 nm axially ($z$) was determined by Fourier ring correlation.

**Reporting summary**. Further information on research design is available in the Nature Research Reporting Summary linked to this article.

## Data availability
All relevant data supporting the findings of this study are available from the corresponding authors upon reasonable request. Source data are provided with this paper.

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

## Acknowledgements

The component of this study carried out at Stanford was supported by the NIMH (MH052804 to T.C.S), and the component carried out at Bioland Laboratory was supported by the National Natural Science Foundation of China (31970914 to F.L.).

## Author contributions

Conceptualization, F.L. and T.C.S.; Methodology, F.L., A.S., S.M., T.C.S.; Investigation, F.L., A.S.; Writing—Original Draft, F.L., A.S., and T.C.S.; Writing—Review & Editing, F.L., A.S., S.M., and T.C.S.

## Competing interests

The authors declare no competing interests.
