## [Peer Review File · Nature Communications]

REVIEWER COMMENTS

Reviewer #2 (Remarks to the Author):

In this study, Luo et al. analyze the role of Neurexin (Nrxn) for presynaptic GABAB-receptor regulation and localization at four central mammalian synapses. A previously reported central function of presynaptic GABAB-receptors is to inhibit presynaptic Ca²⁺-channels. The authors use state of the art electrophysiological analysis and the previously generated triple Nrxn123 conditional KO mice (a.k.a pan-neurexin KO mice) to test whether their pan-neurexin deletion affects the GABAB-receptor-dependent presynaptic inhibition of neurotransmitter release. The authors show that at all four central synapse types (two inhibitory and two excitatory), GABAB-receptor activation, via SKF-97541 treatment, in control samples and the pan-neurexin KO animals decreased evoked EPSCs, followed by an increase in paired-pulse ratio (PPR) and a reduced release probability. Interestingly, however, in pan-neurexin KO mice, GABAB-receptor activation produced a much smaller suppression of release probability, leading to a decreased inhibition of postsynaptic responses, a less pronounced elevation in the PPRs, and an enhancement of the coefficient of variation of postsynaptic currents. Thus, they conclude that Nrxns counteract an inhibition of the GABAB-receptor activation upon Ca²⁺ channels at the presynaptic terminus. In addition, the authors shows that the same pan-neurexin deletion has unimpaired gating and current density of presynaptic Ca²⁺ channels, although the application of the GABAB-receptor activation at mutant synapses shows a reduced efficacy in blocking Ca²⁺-currents in the pan-neurexin mutant animals as compared to the non-mutant condition, suggesting Nrxn deletion influences the activity of presynaptic GABAB-receptors on presynaptic Ca²⁺-channels. Lastly, they provide evidence for a reduced labeling intensity of GABAB1&B2-receptors at presynaptic VGLUT positive sites, and infer that Nrxn is involved in the expression/localization/clustering of GABAB-receptors.

This is an interesting study that using sophisticated electrophysiology at a wide selection of inhibitory and excitatory central synapses demonstrates a similar influence of loss of Nrxns on presynaptic GABAB-receptors. Also given the importance of GABAB-receptor signaling for the control of circuit function, the message is relevant to a large spectrum of scientists beyond the outspoken active zone experts.

This said, their study in my eyes would further profit from a direct co-staining of Ca²⁺-channels with Nrxns, preferably using super resolution light and even electron microscopy. Thus, they could further qualify their claim concerning an organisational “nanopattering” role of Nrxns for GABAB-receptors and Ca²⁺-channel clustering at active zone release sites.

Specific points:

1. Figure 3: for my view: the GABAB1&B2-receptors staining does just not look significantly reduced as the quantitative analysis suggests. Maybe they could choose a more representative image here.

2. The authors showed in a parallel recent study that the same pan-neurexin deletion, also performed at calyx synapses, has unimpaired gating and current density of presynaptic Ca²⁺ channels, although the efficiency of release is affected, and that the apparent sensitivity of evoked release to Ca²⁺ is right-shifted. In this study they also show a similar effect of the pan-neuronal neurexin ablation causing a diffusion of Ca²⁺ channel away from the active release sites slots (F. Jou et al, 2020-EMBO J (2020)39:e103208). Thus, the pan-neurexin ablation causes diffusion of Ca²⁺ channel and GABAB-receptors away from presynaptic sites. It would be good they could comparatively discuss these findings more explicitly.

3. It appears to me that it is unclear from their results, at least in the moment, whether the Nrnx mediated regulation of a GABAB-receptors is uncoupled mechanism uncoupled from its regulation of Ca²⁺ channel or, as proposed, whether Nrnx mediated regulation of GABAB-receptors is Nrnx's pathway to regulation of presynaptic Ca²⁺ channel abundance and release. The data on Ca²⁺ channel regulation, from the author's recent EMBO paper, and here GABAB-receptor via Nrnxns have to be consolidated further to strengthen the mechanistic claim of Nrnx mediated modulation of Ca²⁺ channels by GABAB-receptor, in this paper. An N & P/Q-type Ca²⁺ channel co-staining with GABAB1&B2-receptors in pan-neurexin deletion background might be a way to link the claims of Fig 2 and 3 better.

4. In addition these stainings applied onto samples treated + and -SKF are required to show the concomitant reduction and physical reduced levels of both presynaptic proteins in the KO at the same synapses.

Reviewer #3 (Remarks to the Author):

The study by Luo et al demonstrates that in the absence of all neurexins, presynaptic inhibition via GABA_A receptors is severely disturbed in different types of synapses in mouse brain.

The design of this study is very strong, with sparse in vivo inactivation of all three neurexin genes, a selective agonist, selective optogenetic activation and patch clamp analysis of synaptic transmission in four populations of synapses. The observed loss of presynaptic inhibition is major and convincing, with an almost complete loss of GABA_A induced presynaptic inhibition in several synapses in the absence of neurexins. Hence, the study reaches an important novel conclusion with the best possible approaches.

This study builds on previous observations indicating that neurexins are presynaptic organizers of Ca²⁺- and K⁺-channels and now indicates that in the absence of neurexins, GABA_A receptors are functionally uncoupled from their effector process (synaptic transmission), and also physically disconnected (reduced staining intensity in synapses). The authors did not investigate how neurexins link GABA_A receptors to the synapse and to the regulation of synaptic transmission (and which Nrnx genes and splice variants do so), but instead made major efforts to demonstrate that this link applies to different types of synapses. Given the diversity of synapses in the mammalian brain, this is an important new insight.

While the functional evidence is robust, the localization studies are less so and do not exploit approaches with the best possible resolution. The relatively weak evidence for reduced synaptic localization of GABA receptors makes it difficult to rule out that the observed loss of presynaptic inhibition can be explained by disorganized Ca²⁺-channels only. Maybe GABA receptor activation inhibits Ca²⁺-channels to the same extent in both experimental groups, but the inhibition is not observed nearly to the same extent because Ca²⁺-channels are already poorly coupled to exocytosis in the absence of Nrnx123. Finally, most data sets are based on small numbers of observations (3 mice per group) and statistical significance may be inflated by assuming the number of neurons/slices as the number of independent observations.

Major point:

The conclusion that the distribution of GABA_B receptors is altered in the absence of all neurexins is based on normalized intensity of an indirect immunofluorescence signal. No negative controls are presented (minus primary ab, GABA_B-KO sections), no stainings for other synaptic proteins (except vGluT1) and no attempts to calibrate the signals. In the raw data presented (Fig 3A,C) it is impossible to see differences. The images appear over-exposed. How can a difference in receptor density be deduced from these signals? In fact, if anything, the distribution of vGluT1 appears less patchy in the Nr_x123TKO (Fig 3A). Finally, regular immunofluorescence as used here does not have the resolution to observe the most relevant differences between active zone area and the rest of the synaptic membrane. It would be more convincing to demonstrate more clear differences using a more quantitative analysis of immunofluorescence, or superresolution imaging (or immunoEM) and by biochemical techniques (GABA_B receptor levels in synaptosomes upon pan-neuronal inactivation of Nr_x123).

It also remains uncertain how the modest, semi-quantitative reduction in GABA_B receptors can explain the strongly impaired presynaptic inhibition.

The authors' claim that altered distribution and functionality of GABA_B receptors explains the impaired presynaptic inhibition can be tested directly at least in calyx of Held synapses by measuring the Ca²⁺ current density before and after SKF in both experimental groups. Only if before/after SKF is really different between the groups, the conclusion is justified that an altered distribution and functionality of GABA_B receptors explains the impaired presynaptic inhibition.

At the very least, the author should discuss the possibility that the reduced presynaptic inhibition may be explained by disorganized Ca²⁺-channels only and also point out that the presumed reduction in GABA_B receptor density is different from the previously observed effect on Ca²⁺-channels, where the synaptic density is not different, only their functional organization.

Minor points:

-Fig 3: specify imaging methodology in the legend (confocal microscopy)

-Labelling of Fig4e: is the EPSC amplitude really identical between the groups or are these normalized data?

-Line 258: synapse -> synapses

Point-by-point response to the reviewers' comments

We thank the reviewers for their positive and constructive comments on our work. To address major questions raised by both reviewers, we have performed two-color super-resolution microscopy imaging to explore the distribution of presynaptic GABAB_B-receptors. The data are included in a new figure (Fig. 4 in the current revision). We have accordingly revised the manuscript with tracked changes. We addressed specific comments in our point-by-point responses below. The reviewer's comments are cited in full in *italic* and our responses in **bold**.

Reviewer #2 (Remarks to the Author):

In this study, Luo et al. analyze the role of Neurexin (Nrxn) for presynaptic GABAB-receptor regulation and localization at four central mammalian synapses. A previously reported central function of presynaptic GABAB-receptors is to inhibit presynaptic Ca²⁺-channels. The authors use state of the art electrophysiological analysis and the previously generated triple Nrxn123 conditional KO mice (a.k.a pan-neurexin KO mice) to test whether their pan-neurexin deletion affects the GABAB-receptor-dependent presynaptic inhibition of neurotransmitter release. The authors show that at all four central synapse types (two inhibitory and two excitatory), GABAB-receptor activation, via SKF-97541 treatment, in control samples and the pan-neurexin KO animals decreased evoked EPSCs, followed by an increase in paired-pulse ratio (PPR) and a reduced release probability. Interestingly, however, in pan-neurexin KO mice, GABAB-receptor activation produced a much smaller suppression of release probability, leading to a decreased inhibition of postsynaptic responses, a less pronounced elevation in the PPRs, and an enhancement of the coefficient of variation of postsynaptic currents. Thus, they conclude that Nrxns counteract an inhibition of the GABAB-receptor activation upon Ca²⁺ channels at the presynaptic terminus. In addition, the authors shows that the same pan-neurexin deletion has unimpaired gating and current density of presynaptic Ca²⁺ channels, although the application of the GABAB-receptor activation at mutant synapses shows a reduced efficacy in blocking Ca²⁺-currents in the pan-neurexin mutant animals as compared to the non-mutant condition, suggesting Nrxn deletion influences the activity of presynaptic GABAB-receptors on presynaptic Ca²⁺-channels. Lastly, they provide evidence for a reduced labeling intensity of GABAB1&B2-receptors at presynaptic VGLUT positive sites, and infer that Nrxn is involved in the expression/localization/clustering of GABAB-receptors.

This is an interesting study that using sophisticated electrophysiology at a wide selection of inhibitory and excitatory central synapses demonstrates a similar influence of loss of Nrxns on presynaptic GABAB-receptors. Also given the importance of GABAB-receptor signaling for the control of circuit function, the message is relevant to a large spectrum of scientists beyond the outspoken active zone experts.

This said, their study in my eyes would further profit from a direct co-staining of Ca²⁺-channels with Nrxns, preferably using super resolution light and even electron microscopy. Thus, they could further qualify their claim concerning an organisational "nanopattering" role of Nrxns for GABAB-receptors and Ca²⁺-channel clustering at active zone release sites.

We appreciate the reviewer's positive assessment of our manuscript. We totally agree that imaging the spatial distribution of GABA_B-receptors with super resolution microscopy should be helpful in deciphering how disorganized GABA_B receptors might impair their functional regulation of Ca²⁺ channels. We thus performed dSTORM imaging to explore the relevant distribution of GABA_B receptors within presynaptic terminals, in reference to the distribution of postsynaptic Homer1, in WT and Nrnx mice. We included the new dataset in a new figure (Fig. 4) and addressed the reviewer's question in our response to specific points 2-3 below.

Specific points:

1. *Figure 3: for my view: the GABAB1&B2-receptors staining does just not look significantly reduced as the quantitative analysis suggests. Maybe they could choose a more representative image here.*

GABA_B receptors are universally expressed in both presynaptic and postsynaptic compartments of neurons as well as in astrocytes. The antibodies against GABA_{B1} or GABA_{B2} respectively should label all receptors in the tissue. When we compared the overall staining of GABA_{B1} or GABA_{B2} in the tissue, we found no difference in control and Nrnx TKO mice. However, when we restricted our quantitative analysis on the presynaptic terminal, by analyzing the signal of GABA_{B1} or GABA_{B2} receptors colocalized with VGLUT1, a reliable presynaptic marker, we found a modest but significant reduction in the intensity of GABA_{B1} or GABA_{B2} staining in Nrnx TKO mice vs. control mice. We clarified the issue in both text and figure legend.

2. *The authors showed in a parallel recent study that the same pan-neurexin deletion, also performed at calyx synapses, has unimpaired gating and current density of presynaptic Ca²⁺ channels, although the efficiency of release is affected, and that the apparent sensitivity of evoked release to Ca²⁺ is right-shifted. In this study they also show a similar effect of the pan-neuronal neurexin ablation causing a diffusion of Ca²⁺ channel away from the active release sites slots (F. Luo et al, 2020-EMBO J (2020)39:e103208). Thus, the pan-neurexin ablation causes diffusion of Ca²⁺ channel and GABAB-receptors away from presynaptic sites. It would be good they could comparatively discuss these findings more explicitly.*

Agree. We have explicitly discussed how the pan-neurexin deletion impact on the distribution of both Ca²⁺ channels and GABA_B-receptors and incorporated these findings in our understanding of neurexin function as a presynaptic organizer in coordinating various signaling molecules at the active zones. See the highlighted part in page 11: "Two potential mechanisms may underlie the effect of pan-neurexin deletion on GABA_B-receptor inhibition of Ca²⁺ channels and thus neurotransmitter release. First, mis-localization of Ca²⁺ channels in neurexin-deficient synapse³⁴ may simply increase their distance to GABA_B-

receptor cluster and therefore render the inhibition less sensitive. However, we think that this possibility is highly unlikely because pan-neurexin deletion causes no change in Ca^{2+} channels, including their number and distribution in mPFC PV⁺ interneurons³³. Second, pan-neurexin deletion leads to a direct impairment in the distribution or functioning of GABA_B-receptors. Our immunohistochemistry analysis (Fig.3) and dSTORM imaging of GABA_B-receptors (Fig.4) at the calyx of Held terminals reveal a significant reduction in the abundance and localization of presynaptic GABA_B-receptors, suggesting that neurexins play an important role in the abundance and localization of GABA_B-receptors. Moreover, we observe a strong ablation of GABA_B-receptor inhibition on Ca^{2+} channels in both hippocampal and cerebellar PV⁺ interneurons (Fig.6&7). The effects on the abundance and localization of presynaptic GABA_B-receptors appear to be modest, as compared to the strong impact on inhibition of EPSC/IPSC, which thus may warrant an additive role of neurexin in modulating the functioning of GABA_B-receptors. “

3. It appears to me that it is unclear from their results, at least in the moment, whether the Nrnx mediated regulation of a GABA_B-receptors is uncoupled mechanism uncoupled from its regulation of Ca²⁺ channel or, as proposed, whether Nrnx mediated regulation of GABA_B-receptors is Nrnx's pathway to regulation of presynaptic Ca²⁺ channel abundance and release. The data on Ca²⁺ channel regulation, from the author's recent EMBO paper, and here GABA_B-receptor via Nrnxns have to be consolidated further to strengthen the mechanistic claim of Nrnx mediated modulation of Ca²⁺ channels by GABA_B-receptor, in this paper. An N & P/Q-type Ca²⁺ channel co-staining with GABA_B1&B2-receptors in pan-neurexin deletion background might be a way to link the claims of Fig 2 and 3 better.

We agree. Imaging the spatial distribution of GABA_B receptors with super resolution microscopy would be helpful in deciphering how neurexins may modulate GABA_B receptors expression and how disorganized GABA_B receptors might impair their functional regulation of Ca^{2+} channels. We have performed dSTORM imaging to explore the relevant distribution of GABA_B receptors within presynaptic terminals in WT and Nrnx mice. These findings provide further support that neurexins are indeed presynaptic active zone organizer in parallel for numerous signaling molecules including Ca^{2+} channels, BK channels, and GABA_B receptors.

4. In addition these stainings applied onto samples treated + and -SKF are required to show the concomitant reduction and physical reduced levels of both presynaptic proteins in the KO at the same synapses.

We disagree. As the SKF effect could be rapidly reversed on both WT and Nrnx TKO synapse (extended data Fig.1), the amount of presynaptic GABA_B receptors remained unchanged during such short-period of SKF treatment.

Reviewer #3 (Remarks to the Author):

The study by Luo et al demonstrates that in the absence of all neurexins, presynaptic inhibition via GABA_B receptors is severely disturbed in different types of synapses in mouse brain.

The design of this study is very strong, with sparse in vivo inactivation of all three neurexin genes, a selective agonist, selective optogenetic activation and patch clamp analysis of synaptic transmission in four populations of synapses. The observed loss of presynaptic inhibition is major and convincing, with an almost complete loss of GABA_B induced presynaptic inhibition in several synapses in the absence of neurexins. Hence, the study reaches an important novel conclusion with the best possible approaches.

This study builds on previous observations indicating that neurexins are presynaptic organizers of Ca²⁺- and K⁺-channels and now indicates that in the absence of neurexins, GABA_B receptors are functionally uncoupled from their effector process (synaptic transmission), and also physically disconnected (reduced staining intensity in synapses). The authors did not investigate how neurexins link GABA_B receptors to the synapse and to the regulation of synaptic transmission (and which Nr_x genes and splice variants do so), but instead made major efforts to demonstrate that this link applies to different types of synapses. Given the diversity of synapses in the mammalian brain, this is an important new insight.

We thank the reviewer's strong comment on our work.

While the functional evidence is robust, the localization studies are less so and do not exploit approaches with the best possible resolution. The relatively weak evidence for reduced synaptic localization of GABA receptors makes it difficult to rule out that the observed loss of presynaptic inhibition can be explained by disorganized Ca²⁺-channels only. Maybe GABA receptor activation inhibits Ca²⁺-channels to the same extent in both experimental groups, but the inhibition is not observed nearly to the same extent because Ca²⁺-channels are already poorly coupled to exocytosis in the absence of Nr_x123. Finally, most data sets are based on small numbers of observations (3 mice per group) and statistical significance may be inflated by assuming the number of neurons/slices as the number of independent observations.

We agree that more imaging of the spatial distribution of GABA_B receptors would strengthen our interpretation and hypothesis that neurexins regulate GABA_B receptor function in parallel with their effects on Ca²⁺ channels. We thus performed dSTORM imaging to explore the relevant distribution of GABA_B receptors within presynaptic terminals, in reference to the distribution of postsynaptic Homer1, in WT and Nr_{xn} mice. We included the new dataset in a new figure (Fig. 4).

We can now rule out the possibility that “loss of presynaptic inhibition can be explained by disorganized Ca²⁺-channels only” for two reasons. First, we clearly observed a significant reduction in SKF inhibition of I_{Ca} directly recorded from the calyx terminals, suggesting the GABA_B-receptor inhibition of Ca²⁺ channels is impaired after the pan-neurexin deletion (Fig.2). Second, because

pan-neurexin deletion causes no change in Ca²⁺ channels including their number and distribution in cerebral PV⁺ interneurons³³, but it leads to a strong ablation of GABA_B-receptor inhibition of Ca²⁺ channels in both hippocampal and cerebellar PV⁺ interneurons (Fig.6&7). We have added the explicit discussion in our revised manuscript in p11.

Major point:

The conclusion that the distribution of GABA_B receptors is altered in the absence of all neurexins is based on normalized intensity of an indirect immunofluorescence signal. No negative controls are presented (minus primary ab, GABA_B-KO sections), no stainings for other synaptic proteins (except VGluT1) and no attempts to calibrate the signals. In the raw data presented (Fig 3A,C) it is impossible to see differences. The images appear over-exposed. How can a difference in receptor density be deduced from these signals? In fact, if anything, the distribution of vGluT1 appears less patchy in the Nr_x123TKO (Fig 3A). Finally, regular immunofluorescence as used here does not have the resolution to observe the most relevant differences between active zone area and the rest of the synaptic membrane. It would be more convincing to demonstrate more clear differences using a more quantitative analysis of immunofluorescence, or superresolution imaging (or immunoEM) and by biochemical techniques (GABA_B receptor levels in synaptosomes upon pan-neuronal inactivation of Nr_x123).

GABA_B receptors are universally expressed in both presynaptic and postsynaptic compartments of neurons as well as in astrocytes. The antibodies against GABA_{B1} or GABA_{B2} respectively should label all receptors in the tissue. When we compared the overall staining of GABA_{B1} or GABA_{B2} in the tissue, we found no difference in control and Nr_{xn} TKO mice. However, when we restricted our quantitative analysis on the presynaptic terminal, by analyzing the signal of GABA_{B1} or GABA_{B2} receptors colocalized with VGluT1, a reliable presynaptic marker, we found a modest but significant reduction in the intensity of GABA_{B1} or GABA_{B2} staining in Nr_{xn} TKO mice vs. control mice. We clarified the issue in both text and figure legend.

We fully agree that more imaging of the spatial distribution of GABA_B receptors at super resolution would strengthen our interpretation and hypothesis that neurexins regulate GABA_B receptor function in parallel with their effects on Ca²⁺ channels. We have used dSTORM imaging technique to explore the relevant distribution of GABA_B receptors within presynaptic terminals in WT and Nr_{xn} mice in relevance to the distribution of postsynaptic marker Homer 1. Refer to our response below and to specific points 2-3 in reviewer 2 comments.

It also remains uncertain how the modest, semi-quantitative reduction in GABA_B receptors can explain the strongly impaired presynaptic inhibition.

The authors' claim that altered distribution and functionality of GABA_B receptors explains the impaired presynaptic inhibition can be tested directly at least in calyx of Held synapses by

measuring the Ca²⁺ current density before and after SKF in both experimental groups. Only if before/after SKF is really different between the groups, the conclusion is justified that an altered distribution and functionality of GABA_B receptors explains the impaired presynaptic inhibition. At the very least, the author should discuss the possibility that the reduced presynaptic inhibition may be explained by disorganized Ca²⁺-channels only and also point out that the presumed reduction in GABA_B receptor density is different from the previously observed effect on Ca²⁺-channels, where the synaptic density is not different, only their functional organization.

Agree. We have added the explicit discussion on the potential mechanisms how neurexins may impact on both Ca²⁺ channels and GABA_B-receptors function. See the highlighted part in page 11: “Two potential mechanisms may underlie the effect of pan-neurexin deletion on GABA_B-receptor inhibition of Ca²⁺ channels and thus neurotransmitter release. First, mis-localization of Ca²⁺ channels in neurexin-deficient synapse³⁴ may simply increase their distance to GABA_B-receptor cluster and therefore render the inhibition less sensitive. However, we think that this possibility is highly unlikely because pan-neurexin deletion causes no change in Ca²⁺ channels, including their number and distribution in mPFC PV⁺ interneurons³³. Second, pan-neurexin deletion leads to a direct impairment in the distribution or functioning of GABA_B-receptors. Our immunohistochemistry analysis (Fig.3) and dSTORM imaging of GABA_B-receptors (Fig.4) at the calyx of Held terminals reveal a significant reduction in the abundance and localization of presynaptic GABA_B-receptors, suggesting that neurexins play an important role in the abundance and localization of GABA_B-receptors. Moreover, we observe a strong ablation of GABA_B-receptor inhibition on Ca²⁺ channels in both hippocampal and cerebellar PV⁺ interneurons (Fig.6&7). The effects on the abundance and localization of presynaptic GABA_B-receptors appear to be modest, as compared to the strong impact on inhibition of EPSC/IPSC, which thus may warrant an additive role of neurexin in modulating the functioning of GABA_B-receptors. “

Minor points:

-Fig 3: specify imaging methodology in the legend (confocal microscopy)

As suggested, we updated the figure legend with specific methodology.

-Labelling of Fig4e: is the EPSC amplitude really identical between the groups or are these normalized data?

In these recordings, we adjust the stimulation intensity to have similar amplitude in the first response, in order to measure the effect of SKF and the paired pulse ratio. The amplitude of the first response is therefore not reflecting changes in release probability between Δcre and cre injected mice.

-Line 258: *synapse* -> *synapses*

Changed.

REVIEWER COMMENTS

Reviewer #2 (Remarks to the Author):

I am largely satisfied with their revision and recommend publication.

Reviewer #3 (Remarks to the Author):

the authors have adequately dealt with the issues raised. The two major issues were addressed with new, convincing experiments and with new text in the Discussion.

Point-by-point response to the reviewers' comments

We are happy that we have addressed all major concerns raised by the reviewers. We highly appreciate their positive support and constructive comments on our work. We addressed specific comments in our point-by-point responses below. The reviewer's comments are cited in full in *italic* and our responses in **bold**.

REVIEWERS' COMMENTS

Reviewer #2 (Remarks to the Author):

I am largely satisfied with their revision and recommend publication.

Thank the reviewer for the strong support during the reviewing process.

Reviewer #3 (Remarks to the Author):

the authors have adequately dealt with the issues raised. The two major issues were addressed with new, convincing experiments and with new text in the Discussion.

We are happy to resolve the major issues which clearly strengthened our paper. Thanks for the positive comments.